# Electrocatalyzed direct arene alkenylations without directing groups for selective late-stage drug diversification

Zhipeng Lin[1,8], Uttam Dhawa[1,8], Xiaoyan Hou[1,8], Max Surke[1], Binbin Yuan[1], Shu-Wen Li[2], Yan-Cheng Liou[1], Magnus J. Johansson[3,4], Li-Cheng Xu[2], Chen-Hang Chao[2], Xin Hong[2,5,6] ✉ & Lutz Ackermann[1,7] ✉

Electrooxidation has emerged as an increasingly viable platform in molecular syntheses that can avoid stoichiometric chemical redox agents. Despite major progress in electrochemical C–H activations, these arene functionalizations generally require directing groups to enable the C–H activation. The installation and removal of these directing groups call for additional synthesis steps, which jeopardizes the inherent efficacy of the electrochemical C–H activation approach, leading to undesired waste with reduced step and atom economy. In sharp contrast, herein we present palladium-electrochemical C–H olefinations of simple arenes devoid of exogenous directing groups. The robust electrocatalysis protocol proved amenable to a wide range of both electron-rich and electron-deficient arenes under exceedingly mild reaction conditions, avoiding chemical oxidants. This study points to an interesting approach of two electrochemical transformations for the success of outstanding levels of position-selectivities in direct olefinations of electron-rich anisoles. A physical organic parameter-based machine learning model was developed to predict position-selectivity in electrochemical C–H olefinations. Furthermore, late-stage functionalizations set the stage for the direct C–H olefinations of structurally complex pharmaceutically relevant compounds, thereby avoiding protection and directing group manipulations.

In recent years, molecular electro-organic synthesis has surfaced as a uniquely effective toolbars for sustainable organic syntheses[1–4]. Despite indisputable progress[5–11] by the merger of electrosynthesis and transition metal catalysis, electrochemical C–H activation[12–15] has been largely restricted to the use of directing groups (DG)[16] for the C–H functionalization. These DGs require additional steps for their installation and removal, contrasting the inherent efficacy of the C–H activation[17–21] strategy (Fig. 1a). While the full control of position-selectivity constitutes a major challenge for synthetically useful C–H transformations[22–24], recent advances[25] in catalyst-controlled C–H activation[26–47] have strongly relied on super-stoichiometric amounts of toxic and/or cost-intensive sacrificial chemical oxidants (Fig. 1b).

[1]Wöhler Research Institute for Sustainable Chemistry (WISCh), Georg-August-Universität Göttingen, Göttingen, Germany. [2]Center of Chemistry for Frontier Technologies, Department of Chemistry, State Key Laboratory of Clean Energy Utilization, Zhejiang University, Hangzhou, China. [3]Medicinal Chemistry, Research and Early Development, Cardiovascular, Renal and Metabolism (CVRM), BioPharmaceuticals R&D, AstraZeneca, Gothenburg, Sweden. [4]Department of Organic Chemistry, Stockholm University, Stockholm, Sweden. [5]Beijing National Laboratory for Molecular Sciences, Beijing, PR China. [6]Key Laboratory of Precise Synthesis of Functional Molecules of Zhejiang Province, School of Science, Westlake University, Hangzhou, Zhejiang Province, China. [7]German Centre for Cardiovascular Research (DZHK), Berlin, Germany. [8]These authors contributed equally: Zhipeng Lin, Uttam Dhawa, Xiaoyan Hou. ✉e-mail: hxchem@zju.edu.cn; Lutz.Ackermann@chemie.uni-goettingen.de

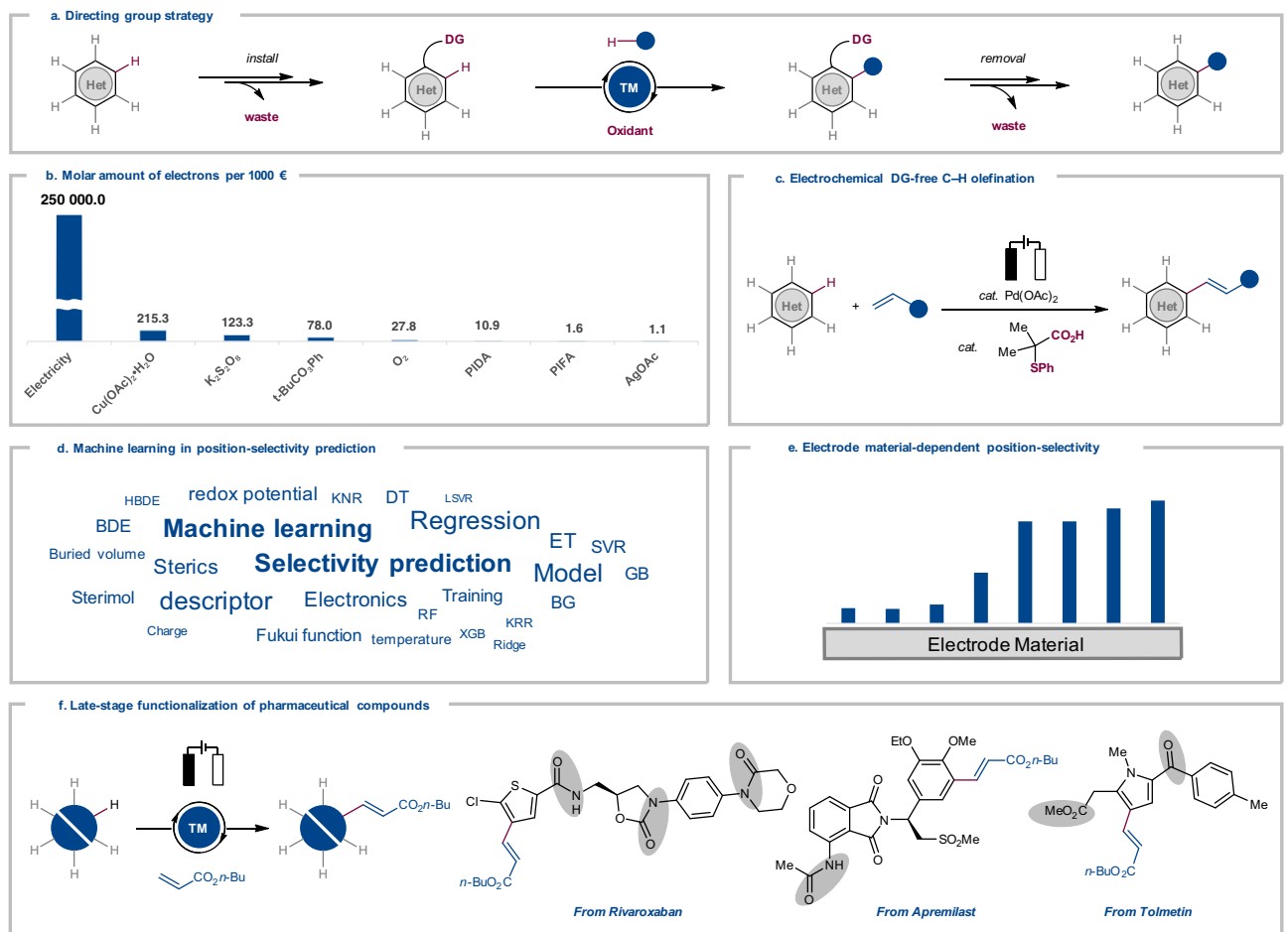

**Fig. 1 | Directing Group (DG)-free Electrochemical C–H Activation. a** Directing group (DG)-assisted oxidative C–H activation by installation and removal of DG. **b** Molar amount of electrons per 1000 euro from electricity and chemical oxidants. PIFA = (bis(trifluoroacetoxy)iodine)benzene. PIDA = (Diacetoxyiodo)benzene. **c** Electrochemical DG-free C–H olefination. **d** Machine learning in position-selectivity prediction. **e** Effects of the electrode material onto site-selectivity. **f** Late-stage functionalization of pharmaceutical molecules. (Potential DG was highlighted in gray).

In this work, we disclose exogenous DG-free palladium-electrochemical C–H olefinations at low temperature without strong stoichiometric oxidants (Fig. 1c)[48]. Additionally, our machine learning (ML) modeling[49–54] provide accurate and efficient prediction of the position-selectivity (Fig. 1d). Our approach is characterized by outstanding position-selectivity in olefinations of the electron-rich substrates by the judicious choice of electrode material (Fig. 1e). Notably, this strategy exhibits high functional group tolerance and unique position-selectivity undisturbed by potential DG, providing straightforward route for the late-stage C–H functionalization of structurally complex molecules of relevance to drug discovery and chemical biology (Fig. 1f).

## Results

We initiated our studies towards the non-directed electrochemical C–H activation by evaluating several *N*-type ligands[27,28], (Supplementary Table 5). We were indeed delighted to observe electrocatalytic activity for *o*-xylene (**1h**) with 2-hydroxy-3-(trifluoromethyl)pyridine (**L1**) as the ligand, notably in the absence of chemical oxidants. Variation of the pyridone resulted in an increased efficacy with ligand **L3**. Protected amino acids **L5**–**L6** and 4,5-diazafluoren-9-one (**L7**) afforded inferior results in the electrocatalysis. Next, a representative set of sulfur-based ligands **L8**–**L12** was tested[40–42,46], and we found an almost quantitative conversion to the olefination product **11** with *S,O*-ligand **L12** by the electrocatalysis.

While altering the stoichiometry did not have considerable influence on the efficacy (Supplementary Table 7, entry 2), the product **11** was not observed when *n*-Bu₄NOAc was used as the supporting electrolyte (entry 3). Likewise, a solvent mixture of TFE and AcOH failed to provide the olefinated product **11** (entry 4). The electrocatalysis occurred efficiently in the absence of 1,4-benzoquinone (BQ), while catalytic amounts of BQ, working as redox mediator in our catalytic system rather than chemical oxidant, is suggested to prevent the aggregation of the palladium catalyst, thereby improving the catalyst's performance (entry 5). Control experiments revealed the necessity of the palladium catalyst, the ligand and the electricity for the DG-free electrochemical C–H activation (entry 6–8). Importantly, scaling up allows to reduce the arene equivalent without any decrease of its efficacy (Supplementary Table 10, entry 1). Thus, the electrolysis for 48 h with 1.5 equiv. of arene and 1 equiv. of alkene along with catalytic amounts of Pd(OAc)₂, ligand **L12**, and BQ, followed by addition of sodium acetate, in acetic acid and hexafluoroisopropanol delivers the olefinated product.

With the optimized electrolysis conditions in hand, we tested the robustness of the DG-free electrochemical olefination (Fig. 2a). We were pleased to find that both electron-rich and more challenging electron-poor arenes **1**–**2** delivered the mono-olefinated products in good to excellent yields by the palladium-electrocatalysis. Electron-deficient fluoro- and chloro-benzene **1b**–**1c** provided the mono-olefinated products **5**–**6** in moderate yields. Subsequently, a wide

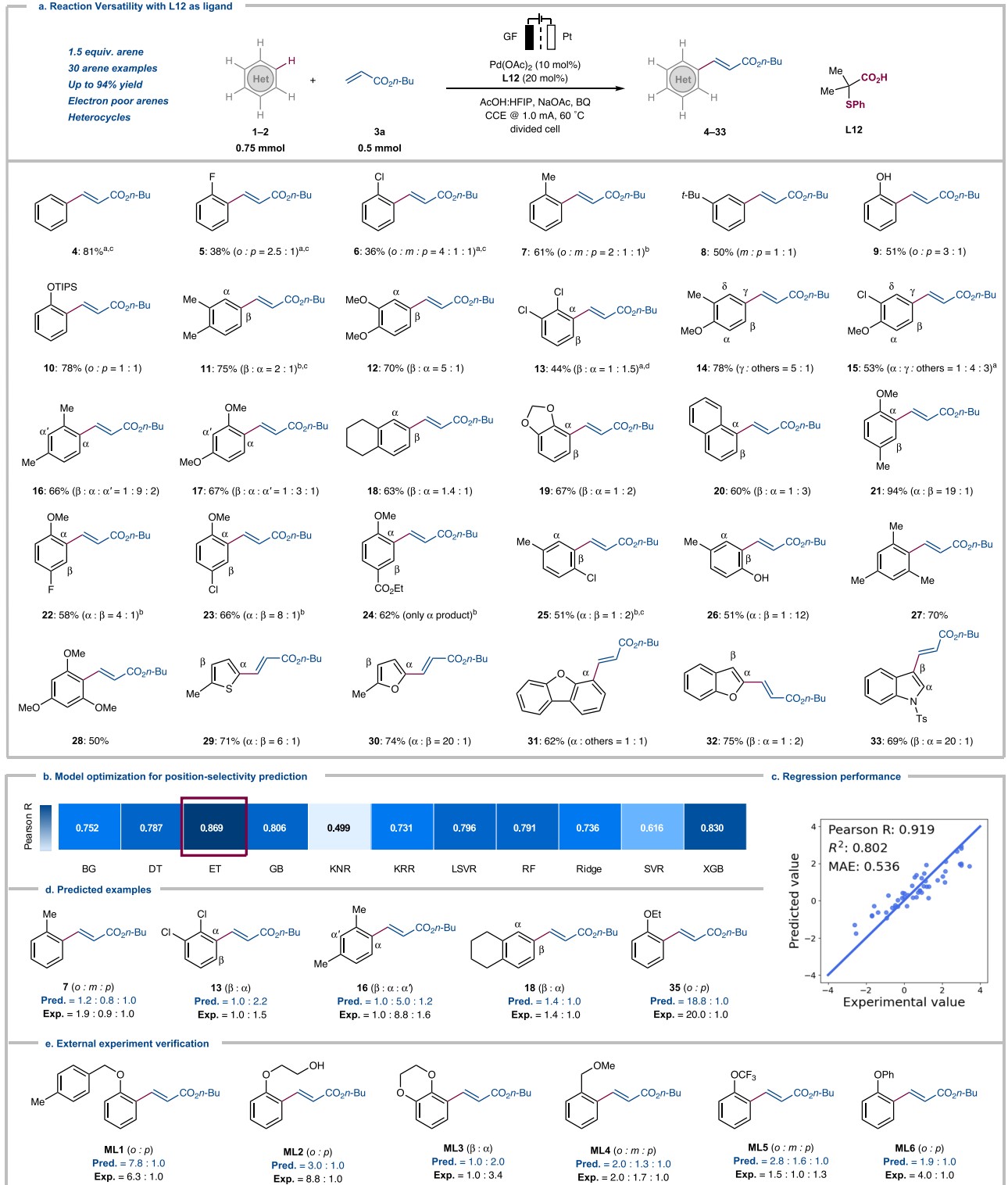

**Fig. 2 | Substrate Scope and Machine Learning.** See supplementary information for reaction details. **a** DG-Free Palladium-Electrochemical C−H Activation. General procedure **C**: divided cell, anodic chamber: **1**−**2** (0.75 mmol), **3a** (0.50 mmol), Pd(OAc)₂ (10 mol %), **L12** (20 mol %), NaOAc (0.20 M), BQ (10 mol %), HFIP:AcOH (1:2); cathodic chamber: NaOAc (0.20 M), BQ (10 mol %), HFIP:AcOH (1:2), constant current at 1.0 mA, 60 °C, 48 h, graphite felt (GF) anode, Pt-plate cathode. ᵃ General

procedure **A**, **1**−**2** (2.0 mmol or 4.0 mmol), **3a** (0.20 mmol). ᵇ **1**−**2** (1.0−1.5 mmol). ᶜ 80 °C. ᵈ 100 °C. **b** Machine learning model selection for position-selectivity prediction (Pearson R). The shadings represent the quality of the regression, the abbreviations embody the machine learning models. **c** Regression performances after descriptor selection. **d** Machine learning predictions for out-of-sample (OOS) examples. **e** Machine learning predictions for external experiment examples.

range of monosubstituted arenes **1d**–**1g** was selectively functionalized, including hydroxyl group OH-free, unprotected phenol **1f** with high yields (**9**). Disubstituted arenes were also examined under the optimized conditions, and for symmetrical disubstituted arenes, the position-selectivity was largely governed by repulsive steric interactions. Dimethoxybenzene (**1i**) yielded mainly the β-olefinated product **12**, while dichlorobenzene (**1j**) gave mainly α-olefinated product **13** instead. 1,2-Disubstituted arenes **1k**–**1l** were predominantly olefinated by the electrooxidation at the γ-position which is *para* to methoxy (**14**–**15**). 1,3-Disubstituted arenes **1m** and **1n** were predominantly olefinated at the α-position (**16**–**17**), while benzodioxole (**1p**) and naphthalene (**1q**) afforded predominantly the α-isomers (**19**–**20**). When *para*-substituted anisoles **1r**–**1u** were tested, the *ortho*-substituted anisole olefinated products **21**–**24** were obtained as the main products. A steric effect was predominant for *para*-chlorotoluene (**1v**) and *p*-cresol (**1w**), since we primarily observed β-isomers **25**–**26**. Furthermore, symmetrical trisubstituted arenes **1x** and **1y** were efficiently converted, delivering the mono-olefinated products **27** and **28**, respectively. It is noteworthy that the robust electrocatalysis was also viable for heteroarenes. Thus, thiophene (**1z**), furan (**2a**), benzofuran (**2b** and **2c**) and indole (**2d**) were efficiently olefinated by the palladium-electrocatalysis to yield the olefinated products **29**–**33** in the absence of chemical oxidants.

Next, in order to accurately predict the site-selectivity, we developed a ML model based on the collected position-selectivity data of all the arenes. A series of physical organic features including buried volume, Sterimol, Fukui function, charge, bond dissociation energies, etc. were applied to encode the involved molecules and enable the ML modeling (Supplementary Table 13). In addition to these site-specific descriptors, the computed redox potential of arenes were also included, considering the importance of electro-oxidation. These molecular descriptors, together with the reaction temperature, created a 28-dimensional encoding for each pair of regioisomeric competing sites, and an array of ML algorithms were evaluated for the regression performance in leave-one-out data splitting (Fig. 2b). The Extra-Trees (ET) model was found to provide the best performance in the position-selectivity prediction, and subsequent feature selection further improves the model's prediction ability while decreasing the complexity of the descriptor space. The resulting ML model revealed a high level of accuracy (Pearson R = 0.919 and mean absolute error (MAE) = 0.536) (Fig. 2c). Feature importance elucidated the determining factors responsible for the regioselectivity prediction, in which the Fukui function of the reacting site emerged as the most crucial parameter (Supplementary Fig. 21). To further validate our model, we tested out-of-sample (OOS) predictions by taking selected arenes out of the training set. The model was revalidated without the access to the regioselectivity data of the selected arenes, and Fig. 2d highlighted a few examples of these excellent OOS predictions. Encouraged by the results, we further tested these predictions experimentally with 6 new arenes. Overall, our models align well with the experimental observations (Fig. 2e), which showed the predictive potential of the model in a rational way to reduce the experimental optimization.

Thereafter, we examined the versatility of the electrocatalysis with regards to anisole derivatives **2** and alkenes **3** (Fig. 3a). Thus, anisole and ethoxy benzene provided high position-selectivity for the *ortho*-functionalized products **34** and **35**. It is noteworthy that these selectivities are complementary to ones previously observed with pyridine-based ligands, which gave *para*-olefinated products as the major isomer[27,31]. Similarly, (benzyloxy)benzene derivatives and propoxybenzene mirrored the selectivity, delivering mono-alkenylated products **36**–**38**. Likewise, a range of alkenes **3** was compatible with the versatile electrochemical conditions, providing an array of olefinated products **39**–**52**. Acrylates **3b**–**3d** including acrylic acid gave outstanding levels of *ortho*-selectivities to afford the products **39**–**41**. Similarly, α,β-unsaturated olefin **3e**–**3g** mirrored this position-

selectivity, providing *ortho*-olefinated products **42**–**44** as the major isomers. α-substituted acrylates (**3h**) were also identified as amenable substrates (**45**). Primary, secondary and tertiary acrylamide (**3j**–**3l**) also well adapted to the superior position-selectivity. Furthermore, the mild nature of the palladium-electrocatalysis manifold allowed for the use of fluorinated alkene **3m**, *NH*-free amino acid derivative **3n** and bio-relevant cholesterol **3o** delivering the predominantly *ortho*-olefinated products **50**–**52**. Here, it is noteworthy that 5.0 equivalents of arene are needed to provide good *ortho*-selectivity.

To delineate the C−H activation elementary step, the potential energy profile was computed at the PBE0-D4/def2-TZVP + SMD(AcOH)//PBE0-D3BJ/def2-SVP level of theory. The formation of the *ortho*-product was found to be kinetically and thermodynamically preferred, with C−H activation barriers of 9.7 kcal mol[-1] (TS(1-2)[*ortho*]) and 10.7 kcal mol[-1] (TS(1-2)[*para*]) respectively (Supplementary Fig. 34a). Non-covalent interactions in the TS(1-2)[*ortho*] further revealed the presence of a weak stabilization interaction between the anisole's methoxy group and the *S,O*-ligand phenyl motif, which contributes to the preferential formation of the *ortho*-product (Fig. 3b).

To further understand the origin of the high position-selectivity with anisoles, we conducted detailed studies (Figs. 3c–g). Here, we observed significant improvement in position-selectivities under the electrochemical conditions compared to reactions with commonly employed chemical oxidants (Fig. 3c). Exploring different electrode materials revealed a remarkable dependence of the position-selectivity on the choice of the material, thereby altering the *ortho/para*-selectivity from 2:1 to 17:1 (Fig. 3d). For such sharp change in selectivities, time-resolved analysis revealed critical insights (Fig. 3e). The ratio of the *ortho/para* selectivity remained constant within the first 12 hours, followed by a considerable change thereafter in favor of the *ortho*-functionalized product. This observation was rationalized by a subsequent selective electrochemical oxidation of the alkene only in the *para*-olefinated product. It is noteworthy that the second oxidation occurred selectively after the alkene **3a** was fully consumed (Supplementary Table 21). The independently prepared *para*-olefinated product **40** was subjected to the standard electrochemical conditions, followed by acetoxylation, resulting in the formation of diacetate **40'** (Fig. 3f). Control experiments showed the essential role of the electricity for the two-fold electrooxidation. CV studies confirmed that the *para*-olefinated product is more labile to be oxidized than the *ortho*-olefinated product, and were well aligned with our experimental observation for electrode material dependence on position-selectivity (Fig. 3g & Supplementary Fig. 12). In contrast, a representative set of commonly employed chemical oxidants were tested such as BQ, AgOAc, K₂S₂O₈, PIDA or TBHP for selective oxidation of the products (Supplementary Table 19). The results were unsatisfactory, highlighting crucial and unique role of electricity in the selective oxidation process.

Finally, the unique power of the DG-free electrocatalysis was exploited for the late-stage functionalization (LSF) of biorelevant drug molecules (Fig. 4)[19–21,47]. C−H olefination of fenofibrate proceeded efficiently to afford the olefinated product **54**. Tolmetin was selectively functionalized at the pyrrole ring to yield the mono-olefinated product **55** in 74% yield. Rivaroxaban was olefinated by the palladium-electrocatalysis to afford **56**. Bezafibrate was selectively converted to product **57**. At the same time, gemfibrozil was effectively converted into the corresponding alkenylated products **58**. Moreover, apremilast was transformed into two separable products **59**. Indomethacin was selectively olefinated to afford **60** in 88% yield at both *ortho*-positions of the anisole. Under the chemical oxidant-free electrocatalysis, naproxen afforded olefins **61**. Ibuprofen was functionalized to provide **62** and ester derivative of estrone were efficiently alkenylated at the *ortho*-position to afford **63**. Likewise, derivatives of ciprofibrate and etodolac provided the olefinated products **64** and **65**. Natural products like khellin and trioxsalen gave single products **66** and **67** under

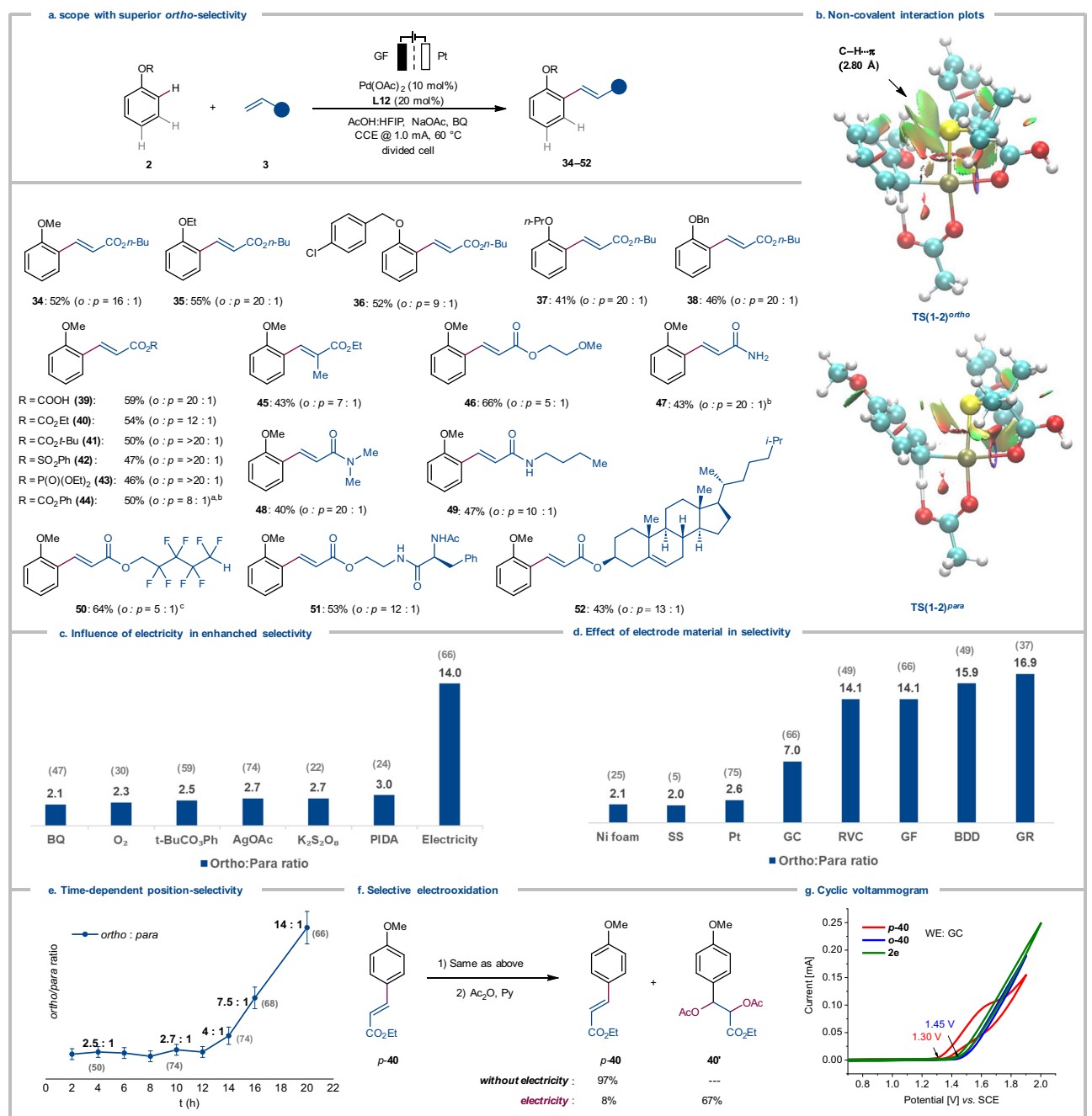

**Fig. 3 | Mechanistic Study for Superior *Ortho*-selectivity.** See supplementary information for reaction details. **a** Scope for anisole and olefin. General procedure **A**: divided cell, anodic chamber: **2** (1.0 mmol), **3** (0.20 mmol), Pd(OAc)₂ (10 mol%), **L12** (20 mol%), NaOAc (0.20 M), BQ (20 mol%), HFIP:AcOH (1:2); cathodic chamber: NaOAc (0.20 M), BQ (20 mol%), HFIP:AcOH (1:2), constant current at 1.0 mA, 60 °C, 20 h, graphite felt (GF) anode, Pt-plate cathode. [a] General procedure **C**, **2** (1.0 mmol), **3** (0.5 mmol). [b] Pt as anode. [c] 80 °C. **b** Non-covalent interaction plots for the TS(1-2)*ortho* and TS(1-2)*para*. **c** Chemical oxidants *vs* Electricity. **d** Variation of anode materials. RVC Reticulated vitreous carbon, GF Graphite felt, BDD Boron doped diamond. GR Graphite rod. **e** *Ortho/para*-selectivity profile. Combined yields of *ortho/para*-**34** were given in the parenthesis. The error bars indicate the possible selectivity fluctuations generated from crude NMR analysis. **f** Selective electrooxidation of *p*-**40**. **g** Cyclic voltammogram for **40**, glass carbon was used as working electrode.

our reaction condition. Etofenprox similarly delivered the olefinated product **68** in 82% yield. Interestingly, vincamine was also tolerated and delivered product **69**. It is noteworthy to mention that our robust electrocatalysis enabled the LSF of complex drug molecules by over-ruling the presence of myriad of strongly coordinating directing groups ranging from ketone to amide and esters.

We have devised a robust and versatile electrochemical direct alkenylation without chemical oxidants and directing groups. The electrochemical olefination was realized by the synergistic cooperation

of electricity, electrode material and a palladium catalyst. A broad variety of alkenes and arenes proved to be compatible with the elec-trooxidative catalysis. A two-fold electrochemical oxidation was uncovered, leading to outstanding levels of selectivity for anisole functionalization. Detailed studies showed the key effect of electricity and the electrode materials in the selectivity control, and machine learning modeling was developed to enable the data-driven position-selectivity prediction. The transformative nature of our strategy was highlighted by late-stage diversifications of bioactive drug molecules

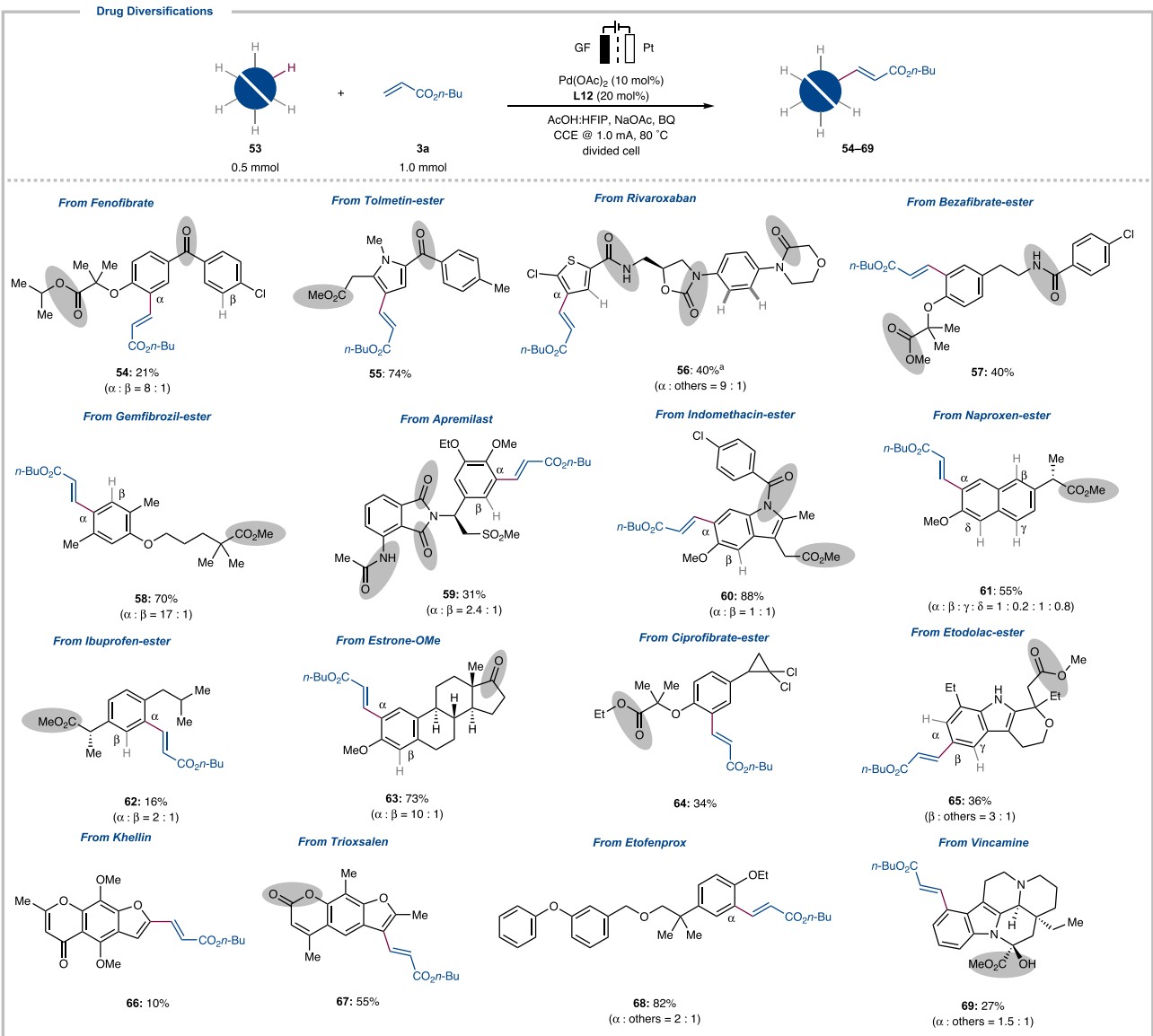

**Fig. 4 | Late-Stage Functionalization of Drug Molecules.** See supporting information for details. Brief reaction condition: **53** (0.5 mmol, 1.0 equiv.), **3a** (1.0 mmol, 2.0 equiv.), divided cell, 80 °C, 48 h. [a] **53** (0.36 mmol, 1.8 equiv.), **3a** (0.2 mmol, 1.0 equiv.). Potential coordinating directing groups are highlighted in gray.

without the installation and removal of any directing groups. Additionally, the electrocatalysis yields molecular hydrogen as the only by-product, representing a synthetically useful anodic oxidation to future green hydrogen technology by the hydrogen evaluation reaction (HER).

## Methods

### General Procedure: Non-directed Electrochemical Olefinations

The electrocatalysis was carried out in a divided cell, equipped with a GF anode and a Pt cathode (10 mm × 15 mm × 0.25 mm). Arenes (0.75 mmol, 1.5 equiv.), acrylates (0.50 mmol, 1.0 equiv.), Pd(OAc)$_2$ (11.3 mg, 10 mol%), ligand (20 mol%), 1,4-benzoquinone (5.4 mg, 10 mol %) and NaOAc (50 mg, 0.20 M) were placed in the anodic chamber and dissolved in AcOH (2.0 mL) and HFIP (1.0 mL); 1,4-benzoquinone (5.4 mg, 10 mol%) and NaOAc (50 mg, 0.20 M) were placed in the cathodic chamber and dissolved in AcOH (2.0 mL) and HFIP (1.0 mL). Galvanostatic electrocatalysis was performed at 60 °C with a current of 1.0 mA and a stirring rate of 500 rpm maintained for 48 h. At ambient temperature, the resulting mixture was diluted with EtOAc (8.0 mL). The GF anode was washed with EtOAc (3 × 10 mL) in an ultrasonic bath. The combined organic phases were loaded on a column and washed with EtOAc (50 mL). The solvents were removed *in vacuo*. Then, NMR was determined by adding CH$_2$Br$_2$ (35.0 µL, 0.50 mmol, 1.0 equiv.) as the standard. The crude mixture was purified by flash column chromatography on silica gel to yield the products.

## Data availability

The authors declare that the data supporting the findings of this study are available within the paper and its Supplementary Information files. Cartesian Coordinates used for DFT calculation were included in Supplementary Data 1. All the involved codes and data in this study are freely available at https://doi.org/10.5281/zenodo.8003927. Source data are provided with this paper and deposited in Zenodo under accession code https://doi.org/10.5281/zenodo.8009809. All other requests for materials and information should be addressed to the corresponding authors.

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

## Acknowledgements

Generous support by the ERC Advanced Grant no.101021358, the DFG (Gottfried Wilhelm Leibniz award, SPP 1807 and SPP 2363), the European Union H2020 research and innovation program under the Marie S. Curie Grant Agreement No 860762 (CHAIR), National Key R&D Program of China (2022YFA1504301, X.H.), the National Natural Science Foundation of China (22122109 and 22271253, X.H.), Zhejiang Provincial Natural Science Foundation of China under Grant No. LDQ23B020002 (X.H.), the Starry Night Science Fund of Zhejiang University Shanghai Institute for Advanced Study (SN-ZJU-SIAS-006, X.H.), Beijing National Laboratory for Molecular Sciences (BNLMS202102, X.H.), the Center of Chemistry for Frontier Technologies and Key Laboratory of Precise Synthesis of Functional Molecules of Zhejiang Province (PSFM 2021-01, X.H.), the State Key Laboratory of Clean Energy Utilization (ZJU-CEU2020007, X.H.), Fundamental Research Funds for the Central Universities (226-2022-00140, 226-2022-00224 and 226-2023-00115, X.H.) and CAS Youth Interdisciplinary Team (JCTD-2021-11, X.H.), the CSC (scholarship to Z.L., X.Y.H., and B.Y.), the DAAD (fellowship to U.D.) and the Ministry of Science and Technology, Taiwan (scholarship 110-2917-I-003-002 to Y.-C.L.) is gratefully acknowledged. Calculations related to M.L. were performed on the high-performance computing system at the Department of Chemistry, Zhejiang University.

## Author contributions

Conceptualization, L.A.; Methodology, U.D., Z.L., L.A., M.J.; Experiment, Z.L., XY.H., M.S., Y.-C.L., and U.D.; DFT, B.Y.; Machine Learning, X.H., S.L., L.X., C.C.; Writing, contributed by all authors; Funding Acquisition, L.A.; Resources, L.A.; Supervision, L.A.

## Funding

## Competing interests

The authors declare no competing interests.
