## [Peer Review File · Nature Communications]

REVIEWERS' COMMENTS

Reviewer #1 (Remarks to the Author):

This revised manuscript addresses most of the remarks and concerns that I formulated on the initial manuscript submitted to Nature. In particular, the authors succeeded to drastically reduce the excess of arene substrate to 1.5 equiv (Fig. 2) and even to use the arene as limiting reagent (Fig. 4), which is much more amenable to late-stage functionalization. These efforts should be commended as the reaction scope was entirely revised with these conditions. They also provided an extended scope and additional insights for the ortho-selective alkenylations (Fig. 3). Finally, they developed a predictive model for the site-selectivity using machine learning (Fig. 2b). This is a great addition, although one would have liked to know which parameters are most critical for the selectivity. Unless there are equal weights of too many parameters in the equation, I suggest that the authors comment on this point in the manuscript. Overall, this is an excellent paper and I recommend acceptance in Nature Communications with the minor revision mentioned above (if applicable).

Reviewer #3 (Remarks to the Author):

Reviewed is an electrocatalytic method for the siteselective

alkenylation of arenes in the absence of a directing group. Substrate scope was found to be very broad enabling late-stage functionalization. Although reviewer 3 raised concerns about the novelty of the approach, the revised version of the manuscript adequately addressed all three reviewers' comments. Therefore, I am confident that this work is now suitable for publication in Nature Communications, given the successful resolution of the issues raised during the review process.

Göttingen
12.06.2023

We warmly thank all of the reviewers for their insightful comments, which have enabled us to significantly improve the quality of our manuscript. Detailed responses to the reviewers' comments are provided below for each point.

Comment 1: *This revised manuscript addresses most of the remarks and concerns that I formulated on the initial manuscript submitted to Nature. In particular, the authors succeeded to drastically reduce the excess of arene substrate to 1.5 equiv (Fig. 2) and even to use the arene as limiting reagent (Fig. 4), which is much more amenable to late-stage functionalization. These efforts should be commended as the reaction scope was entirely revised with these conditions. They also provided an extended scope and additional insights for the ortho-selective alkenylations (Fig. 3). Finally, they developed a predictive model for the site-selectivity using machine learning (Fig. 2b). This is a great addition, although one would have liked to know which parameters are most critical for the selectivity. Unless there are equal weights of too many parameters in the equation, I suggest that the authors comment on this point in the manuscript. Overall, this is an excellent paper and I recommend acceptance in Nature Communications with the minor revision mentioned above (if applicable).*

Response: We are thankful for this valuable suggestion. We have added the key factors for predicting site-selectivity using ML in the manuscript and SI.

“Feature importance elucidated the determining factors responsible for the regioselectivity prediction, in which the Fukui function of the reacting site emerged as the most crucial parameter (Supplementary Figure 16).”

Supplementary Figure 16 Feature importance scores of top-ranking features.

Comment 2: Reviewed is an electrocatalytic method for the site-selective alkenylation of arenes in the absence of a directing group. Substrate scope was found to be very broad enabling late-stage functionalization. Although reviewer 3 raised concerns about the novelty of the approach, the revised version of the manuscript adequately addressed all three reviewers' comments. Therefore, I am confident that this work is now suitable for publication in Nature Communications, given the successful resolution of the issues raised during the review process.

Response: We appreciate this comment.

Best regards,

Lutz Ackermann